# Pharmacokinetics/pharmacodynamics of gamithromycin for treating *Pasteurella multocida* infection in cattle using a tissue cage model

Qingwen Yang[1], Xuesong Liu[2]*, Yongzhi Lv[1], Yushen Li[3]

**1** Laboratory of Veterinary Pharmacology, Department of Animal Science and Technology, Chongqing Three Gorges Vocational College, Chongqing, China, **2** Heilongjiang Province Key Laboratory of Veterinary Drugs, Branch of Animal Husbandry and Veterinary of Heilongjiang Academy of Agricultural Sciences, Qiqihar, China, **3** Qiqihar Senyu Environmental Protection Technology Co., Ltd, Qiqihar, China

* abliuxuesong@yeah.net

## Abstract

In this study, gamithromycin, a long-acting azalide antibiotic recently introduced for bovine respiratory disease (BRD) treatment, was evaluated for its effectiveness against *Pasteurella multocida* using a cattle tissue cage model. Gamithromycin (6 mg/kg) was administered via both intravenous and subcutaneous routes and the gamithromycin contents in sera, transudates, and exudates were measured using HPLC/MS-MS. Non-compartmental methods were utilized for assessing pharmacokinetic parameters and an inhibitory sigmoid $E_{max}$ model determined associations between the pharmacokinetic/pharmacodynamic (PK/PD) indices and antibacterial activity. The area under the 24-h concentration-time curve/minimum inhibitory concentration ($AUC_{0-24h}$/MIC) was found to be an optimal measure of antibacterial activity. The $AUC_{0-24h}$/MIC values over 24 h in sera, transudates, and exudates were 0.27, 0.17, and 0.14, respectively, for bacteriostatic effects, while for bactericidal activity, the $AUC_{0-24h}$/MIC values over 24 h in sera and exudates 3.76 and 5.31, respectively, and for bacterial eradication, the serum value was 18.46. These findings contribute valuable insights into the optimization of gamithromycin dosing regimens for treating respiratory conditions caused by *Pasteurella multocida* in cattle.

## Introduction

*Pasteurella multocida* is a Gram-negative bacillus of the *Pasteurellaceae* family and is one of the main pathogens responsible for bovine respiratory disease (BRD) [1,2]. The incidence of BRD has increased with the rapid development of intensive cattle farming, resulting in significant economic losses in the industry[3]. The disease is usually treated with antibiotics. Therefore, it is important to optimize clinical dosages to achieve maximum efficacy and reduce the production of drug-resistant strains.

**Data availability statement:** All relevant data are within the paper and its Supporting Information files.

**Funding:** This study was supported by the Science and Technology Research Program of Chongqing Municipal Education Commission(Grant No.KJZD-K202303501,KJQN-202203511, KJQN-202203504, KJQN-202303507,KJQN-202403510).

**Competing interests:** The authors have declared that no competing interests exist.

Gamithromycin was developed for preventing and treating BRD resulting from *P. multocida*, *Mannheimia haemolytica,* and *Histophilus somni* infections [4,5]. This second-generation macrolide is a typical azalide, which contains a 15-member carbon ring with a unique alkylated nitrogen in the 7a position. Like other macrolides, gamithromycin achieves bacteriostatic and bactericidal effects by inhibiting RNA-dependent protein synthesis. Due to its wide-ranging antimicrobial spectrum, strong antibacterial action, and rapid clearance, gamithromycin is widely used for treating and preventing BRD[6,7].

The pharmacokinetics (PK)/pharmacodynamics (PD) model (PK/PD) model integrates PK, describing the ADME characteristics of the drug, with PD, focusing on biological effects and associated mechanisms. The model mathematically characterizes relationships between drug levels and their influence over time[8]. The use of PK/PD models can assist assessments of dosages and regimens for achieving the desired therapeutic effect while minimizing resistance. By simulating various dosing scenarios, researchers can identify the most effective and safe dosing strategies[9]. Specific PK parameters, including the area under the curve (AUC) and the maximum concentration ($C_{max}$), and PD parameters, such as the minimal inhibitory concentration (MIC) and time-kill curves, are commonly integrated into PK-PD models. The indices used are the AUC/MIC ratio, the duration of drug level maintenance above the MIC (%T > MIC), and the $C_{max}$/MIC ratio [10]. When using AUC/MIC value for evaluating breakpoints, a time dimension (in hours) is included, often presenting difficulties for reader comprehension and discussion among scientists. However, division of the AUC/MIC value by the time duration produces more easily understandable values. This approach allows for the assessment of a ratio without dimensions, making the computed numerical value more straightforward and clinically interpretable[11].

Pharmacokinetic studies of gamithromycin in various animal species, such as broiler chickens[12], sheep[13], turkeys[14], and pigs[15], have shown that the drug is rapidly absorbed and widely distributed following subcutaneous (s.c.) administration. However, to the best of our knowledge, there is no information on the combination of *in vitro* PK data with *ex vivo* PD values for predicting the various antibacterial activities of gamithromycin in a cattle tissue cage model. Here, the PK/PD values for gamithromycin action on *Pasteurella multocida* in a tissue cage model in cattle after intravenous (i.v.) or subcutaneous (s.c.) injection were investigated. These data represent a resource for optimizing dose regimens in treating BRD caused by *Pasteurella multocida.*

## Materials and methods

### Antibiotics, bacteria and chemicals

Injectable gamithromycin solution (Zactran,ᵒ 150 mg/mL) was purchased from Merial (lot number GYR053CB; Toulouse, France). Gamithromycin standard (purity >98%) was from Tiaozhan Biotechnology (Tianjin, China). After isolation from cattle, *P. multocida* NM-5–7 was identified using a MALDI-TOF/MS system (Axima-Assurance-Shimadzu). Tryptic Soy agar (TSA) and Mueller–Hinton II Broth (MHB) were from

Huankai Microbial Technology (Guangdong, China), defibrinated sheep blood was from Ruite Biological Technology (Guangdong, China), and λ-Carrageenan was purchased from Sigma–Aldrich (lot number BCBP8978V; Shanghai, China).

### Cattle tissue cage design and surgical implantation

Four tissue cages were implanted as previously described[16]. Briefly, golf practice balls (internal diameter, 42 mm; internal volume, 36.1 mL) and 26 holes (diameter, 5 mm) were used as tissue cages. One cage was implanted in each flank, 8–10 cm from the mid-line, followed by infection protection with 10 000 IU/kg penicillin by intramuscular injection for 3 days, accompanied by simultaneous intramuscular injections of aminopyrine for analgesia. When the wounds had healed and the tissue cages were filled with fluid, 1% sterile carrageenan solution (0.5 mL) was injected into one of the tissue cages. After five weeks, transudates and exudates were harvested from the tissue cages.

### Animals and experimental design

Six healthy cattle (mean weight ± SD, 142 ± 18 kg) were randomlyy allocated to two groups (n = 3 per group). The animals received antibiotic-free daily feed, hay, and water *ad libitum*.

The study design was a two-way crossover, with each animal receiving received gamithromycin (6 mg/kg) both i.v. and s.c. In the first stage of the study, the first group were given 6 mg/kg body weight (bw) gamithromycin i.v. *via* the left jugular vein, while the second group were given the same gamithromycin dose s.c. into the neck. During the second part of the study, the treatments of the groups were reversed. The interval between the two stages was 28 days. All protocols were approved by the Animal Use and Care Committee of Chongqing Three Gorges Vocational College (Chongqing, China).

### Sample collection

The procedures were conducted as described previously[17]. Briefly, blood (10 mL) was sampled from the right jugular vein prior to gamithromycin injection and after 5, 10, 15, 30, and 45 min and 1, 2, 3, 6, 9, 12, 24, 48, 72, 96, 120, 144, 168, and 192 h. Transudate and exudate samples (1.5 mL) were obtained before and at 1, 3, 6, 9, 12, 24, 48, 72, 96, 120, 144, 168, and 192 h after gamithromycin administration. All samples were placed in plastic tubes (without anticoagulant) and were protected from sunlight. The samples were centrifuged (4000 × *g*, 4°C, 10 min) and retained at -80°C until analyzed.

### LC-MS/MS determination of gamithromycin concentrations

The levels of gamithromycin in serum, transudate, and exudate samples were evaluated using analyzed using LC-MS/MS, as described previously[16,18]. MS/MS was conducted in the positive electrospray ionization mode. The reaction transitions of gamithromycin monitored included the protonated molecule to product ion transitions of m/z 777.45 > 619.40 for quantification and m/z 777.45 > 157.70 for identification. The gradient elution was performed with mobile phase A consisting of 1% formic acid in water, and acetonitrile as mobile phase B. The details are provided in Table 1.

Table 1. Mobile phase conditions for HPLC/MS/MS of gamithromycin.

| Time (min) | Flow rate (mL/min) | Mobile phase A% | Mobile phase B% |
|---|---|---|---|
| 0 | 250 | 95 | 5 |
| 0.5 | 250 | 60 | 40 |
| 2 | 250 | 60 | 40 |
| 2.5 | 250 | 30 | 70 |
| 4 | 250 | 30 | 70 |
| 4.1 | 250 | 95 | 5 |
| 10 | 250 | 95 | 5 |

**Pharmacokinetic analysis of gamithromycin**

The PK parameters used were elimination half-life ($t_{1/2\beta}$), time of maximum concentration ($T_{max}$), area under concentration-time curve ($AUC_{0-last}$), peak drug concentration ($C_{max}$), mean residence time ($MRT_{0-last}$), the volume of distribution ($V_d$), and clearance (CL) of gamithromycin in the sera, transudates, and exudates, determined by non-compartmental analysis (WinNonlin 5.2.1, Pharsight Corporation, USA) and presented as mean±SD.

*In vitro* **susceptibility studies and time-kill curves**

The gamithromycin MICs against *P. multocida* were determined in Mueller Hinton Broth (MHB), sera, exudates, and transudates using the microdilution techniques of the Clinical and Laboratory Standards Institute[19]. The *Streptococcus pneumoniae* ATCC 49619 served as the MIC quality control strain. The MBC is the lowest concentration that achieves a 99.9% bacterial count reduction. After the determination of the MIC, the *in vitro* time-kill curves were then evaluated as described[20]. Briefly, a 10 µL aliquot of bacterial suspension containing an estimated $10^8$ CFU/mL was exposed to gamithromycin at 0.25×, 0.5×, 1×, 2×, 4×, 8×, 16×, or 32×MIC diluted in MHB and serum (1 mL). The cultures were incubated at 37°C and analyzed at 0, 2, 4, 6, 8, 10, 12 and 24 h post-treatment. One hundred-microliter samples of each culture were serially diluted (10-fold), and 20 µL aliquots were placed on 5% sterile defibrinated sheep blood trypticase soy agar (TSA) for 18–24 h at 37°C, followed by colony counting. The limit of detection (LOD) was 200 CFU/mL.

*Ex vivo* **time-kill curves**

The *ex vivo* time-kill curves were constructed as described [20]. Briefly, 10 µL of bacterial culture in the stationary growth phase was added to 1 mL samples of sera, transudates, and exudates, to about $5×10^5$ CFU/mL and cultivated at 37°C with counting at 0, 2, 5, 8, and 24 h post-treatment. Aliquots (100 µL) of the cultures underwent 10-fold serial dilution and 20 µL of each dilution was placed on 5% sheep blood TSA. Colony numbers were determined after incubation for 18–24 h at 37°C. The LOD was 200 CFU/mL.

**Integration of PK/PD and modeling**

The mean PK parameters were evaluated using WinNonlin software.. The MIC in serum was used for modeling. The PK/PD parameters were determined using the inhibitory effect $E_{max}$ model as follows:

$$E = \text{E}_{max} - \frac{(\text{E}_{max} - \text{E}_0) \times \text{C}_e^{N}}{\text{EC}_{50}^{N} + \text{C}_e^{N}}$$

where $E$ indicates the antibacterial effect, i.e., the changes in $\log_{10}$ CFU/mL in the samples (sera, transudates, and exudates) between the initial $\log_{10}$ CFU/mL and 24 h post-incubation; $E_{max}$ denotes the $\log_{10}$ CFU/mL change in the control samples; $E_0$ represents the maximum antibacterial activity, i.e., $\log_{10}$ CFU/mL changes over a 24-h incubation; $EC_{50}$ indicates the AUC/MIC at 50% of $E_{max}$; $C_e$ denotes the PK/PD parameters (%T>MIC, $C_{max}$/MIC, AUC/MIC), and N represents the Hill coefficient.

**Statistical analyses**

Intergroup comparisons were evaluated by *t*-tests using SPSS software 13.0 and verified by Bonferroni's correction. $P < 0.05$ was considered statistically significant.

# Results

### Validation of HPLCC analytic methods

The retention time of gamithromycin was approximately 6.21 min with no interfering impurity peaks. The chromatograms of blank serum, transudate, and exudate samples are shown in Fig 1, while those of spiked samples are shown in

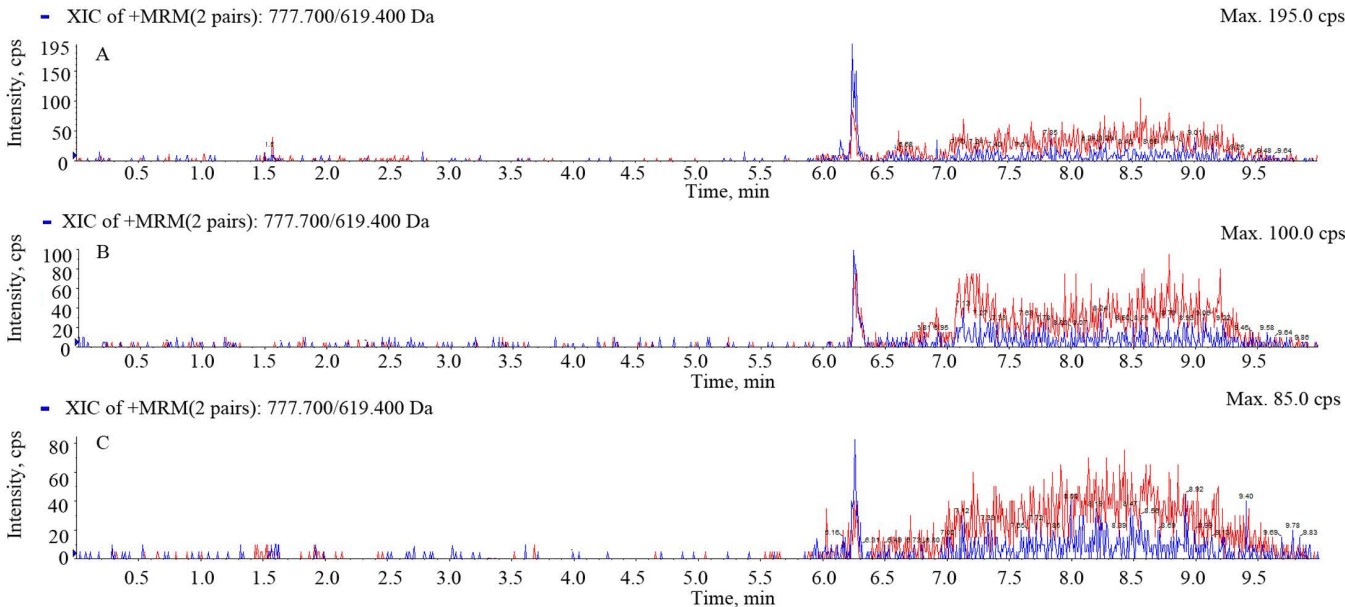

**Fig 1. Chromatograms of blank serum, transudate, and exudate samples.** A, blank serum; B, blank transudate; C, blank exudate.

Fig 2 and those of the separated samples are shown in Fig 3. The LOD and limit of quantification (LOQ) values were 1 and 2 ng/mL, respectively. Recovery of gamithromycin in all samples was over 85%. Variations within and between assays evaluated using the relative standard deviation (%RSD) were all < 10%.

## Pharmacokinetic analysis

The concentration-time curves for gamithromycin in sera, transudates, and exudates after i.v. and s.c. injection of 6 mg/kg bw are illustrated in **Fig 4** and **Fig 5**. The mean concentration-time profiles were determined using non-compartmental models. The main PK parameters of after i.v. administration of gamithromycin in the different samples are shown in **Table 2**, while those following s.c injection are shown in Table 3.

After a single i.v. injection, gamithromycin demonstrated the following pharmacokinetic parameters. In bovine serum, $T_{1/2\beta}$ was 35.27 h, $AUC_{0-last}$ was 5.52 µg·h/mL, MRT was 25.86 h, and CL was 1.07 L/h.kg, while in the transudate, the $C_{max}$ was 0.09 µg/mL, $T_{max}$ was 4.50 h, $T_{1/2\beta}$ was 41.37 h, $AUC_{0-last}$ was 2.65 µg·h/mL, MRT was 55.97 h, and CL was 2.15 L/h.kg, and in the exudate, $C_{max}$ was 0.11 µg/mL, $T_{max}$ was 5.50 h, $T_{1/2\beta}$ was 38.60 h, $AUC_{0-last}$ was 3.73 µg·h/mL, MRT was 53.30 h, and CL was 1.59 L/h.kg.

The parameters after a single s.c. injection were $C_{max}$ of 0.43 µg/mL, $T_{max}$ of 1.00 h, $T_{1/2\beta}$ of 48.30 h, $AUC_{0-last}$ of 6.23 µg·h/mL, MRT of 33.87 h, and CL of 0.95 L/h.kg in serum, $C_{max}$ of 0.04 µg/mL, $T_{max}$ of 9.50 h, $T_{1/2\beta}$ of 63.10 h, $AUC_{0-last}$ of 1.84 µg·h/mL, MRT of 54.87 h, and CL of 2.83 L/h.kg in transudates, and $C_{max}$ of 0.11 µg/mL, $T_{max}$ of 7.00 h, $T_{1/2\beta}$ of 52.74 h, $AUC_{0-last}$ of 2.99 µg·h/mL, MRT of 49.02 h, and CL of 1.85 L/h.kg in exudates. The absolute bioavailability of gamithromycin after subcutaneous injection was 112.95%.

## *In vitro* susceptibility studies and kill curves

Gamithromycin exhibited antimicrobial activity against *P. multocida* strain NM-5–7 with varying efficacy across different media. In MHB, the MIC and minimum bactericidal concentration (MBC) were 0.50 and 1.00 µg/mL, respectively. However, the antibiotic demonstrated enhanced potency in physiological fluids. Specifically, in sera, transudates, and

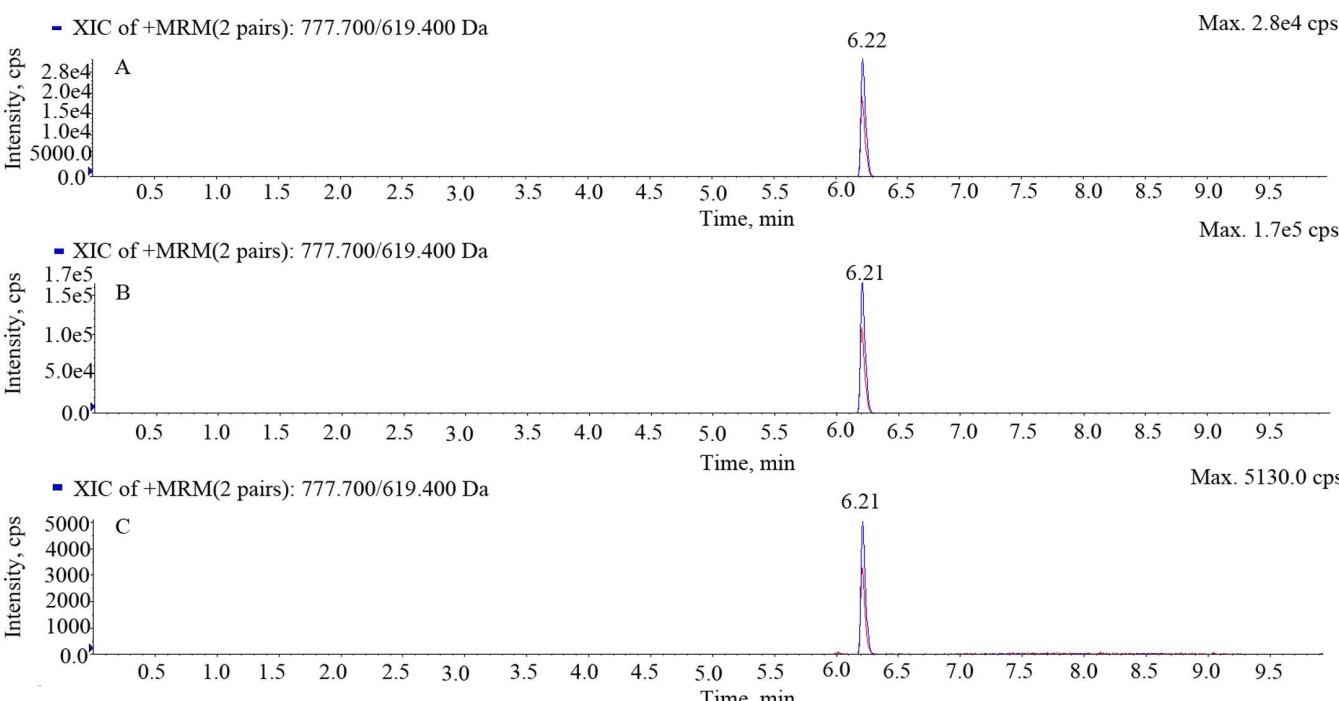

**Fig 2. Chromatograms of spiked serum, transudate, and exudate samples.** A, spiked serum; B spiked transudate; C spiked exudate.

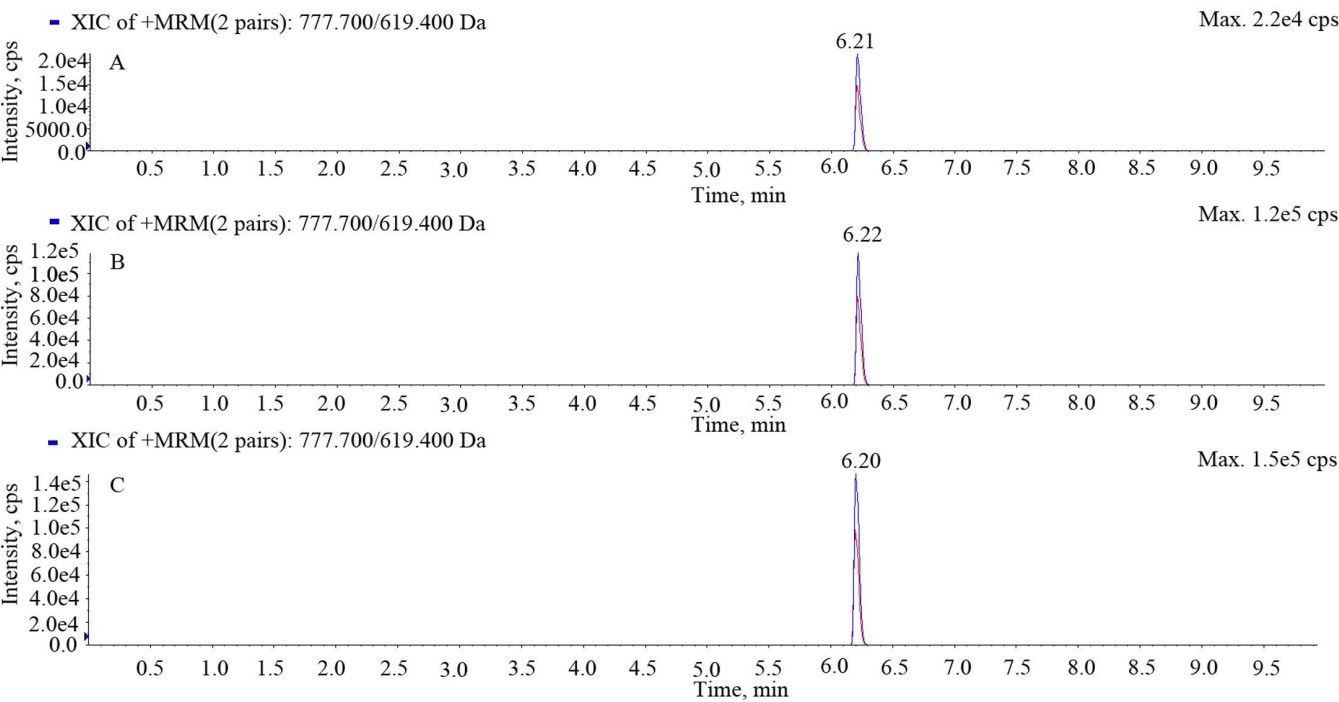

**Fig 3. Chromatograms of separated serum, transudate, and exudate samples.** A, separated serum; B, separated transudate; C, separated exudate.

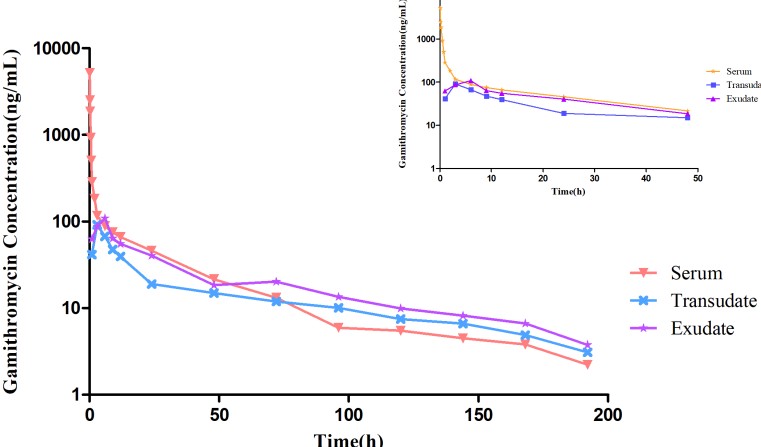

**Fig 4. Concentration-time profiles for gamithromycin in sera, transudates, and exudates after i.v. administration of 6 mg/kg body weight.** The small graph indicates drug concentrations over 48 h following administration.

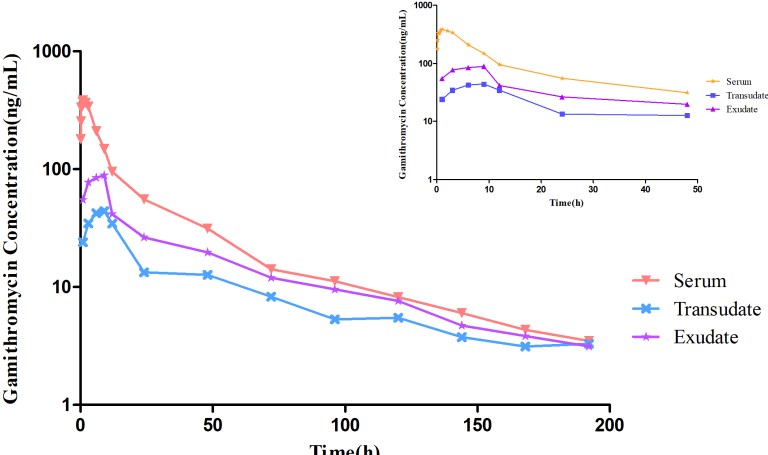

**Fig 5. Concentration-time profiles for gamithromycin in sera, transudates, and exudates after s.c. administration of 6 mg/kg body weight.** The small graph indicates drug concentrations over 48 h following administration.

**Table 2. The pharmacokinetic parameters (mean±SD, n = 6) of gamithromycin in serum, transudate, and exudate after i.v administration of 6 mg/kg.**

| Parameters (units) | Serum | Transudate | Exudate |
|---|---|---|---|
| $C_{max}$ (μg/mL) | — | 0.09 ± 0.02 | 0.11 ± 0.02 |
| $T_{max}$ (h) | — | 4.50 ± 2.29 | 5.50 ± 1.22 |
| $T_{1/2\beta}$ (h) | 35.27 ± 1.79 | 41.37 ± 5.28 | 38.60 ± 2.92 |
| $AUC_{0-last}$ (μg.h/mL) | 5.52 ± 0.54 | 2.65 ± 0.46 | 3.73 ± 0.73 |
| $MRT_{0-last}$ (h) | 25.86 ± 2.48 | 55.97 ± 4.32 | 53.30 ± 6.41 |
| Vd (L/kg) | 54.33 ± 4.55 | — | — |
| CL (L/h.kg) | 1.07 ± 0.13 | 2.15 ± 0.30 | 1.59 ± 0.31 |

**Table 3. The pharmacokinetic parameters (mean±SD, n = 6) of gamithromycin in serum, transudate, and exudate after s.c administration of 6 mg/kg.**

| Parameters (units) | Serum | Transudate | Exudate |
|---|---|---|---|
| $C_{max}$ (µg/mL) | 0.43 ± 0.05 | 0.040 ± 0.01 | 0.11 ± 0.03 |
| $T_{max}$ (h) | 1.00 ± 0.48 | 9.50 ± 2.95 | 7.00 ± 2.45 |
| $T_{1/2\beta}$ (h) | 48.30 ± 6.77 | 64.10 ± 13.14 | 52.74 ± 4.20 |
| $AUC_{0-last}$ (µg.h/mL) | 6.23 ± 0.51 | 1.84 ± 0.16 | 2.99 ± 0.26 |
| $MRT_{0-last}$ (h) | 33.87 ± 2.71 | 54.87 ± 3.51 | 49.02 ± 4.63 |
| Vd (L/kg) | 65.73 ± 11.80 | — | — |
| CL (L/h.kg) | 0.95 ± 0.05 | 2.83 ± 0.23 | 1.85 ± 0.15 |
| F (%) | 112.95 ± 4.03 | – | – |

exudates, the MIC was consistently lower at 0.03 µg/mL, while the MBC was 0.06 µg/mL. This indicates a notably increased antimicrobial effect of gamithromycin in these biological fluids compared to standard laboratory media.

The *in vitro* kill curves of gamithromycin against *P. multocida* NM-5–7 in MHB and serum are presented in Fig 6. The antibacterial trends in broth and serum were similar. The antibacterial activity increased as the drug concentration was raised. At gamithromycin levels below 1 × MIC, there was no inhibitory effect, while at 1 × MIC, there was slight inhibitory activity. Significant bactericidal activity was observed at 2 × MIC, with the rate of killing increasing as the drug concentration rose. At or above 4 × MIC, the bactericidal rate was significantly greater than that at 2 × MIC.

### *Ex vivo* antibacterial action of gamithromycin

The *ex vivo* killing curve of gamithromycin against *P. multocida* NM-5–7 is presented in Fig 7. As illustrated in Fig 7A, serum samples collected during the first 3 hours (0.083, 0.17, 0.25, 0.5, 0.75, 1, 2, and 3 h) after subcutaneous injection exhibited inhibitory activity after 2 h of incubation, while samples obtained at 6 h showed inhibitory activity after 5 h of treatment. After 24 h, serum samples obtained at 24 h displayed inhibitory activity and samples obtained at 0.083, 6, 9, and 12 h showed good bactericidal activity, and those collected at 0.17, 0.25, 0.5, 0.75, 1, 2, and 3 h indicated bacterial clearance. However, samples collected at 48 h or later time points exhibited almost no visible antibacterial activity.

As Fig 7A and 7B indicate, the transudate and exudate samples exhibited similar *ex-vivo* activities against *P. multocida* NM-5–7. In the transudate samples, the gamithromycin transudate samples obtained at 6 and 9 h showed inhibitory activity after 24 h of treatment, while no bactericidal activity was seen in samples from the other time points. Exudate samples

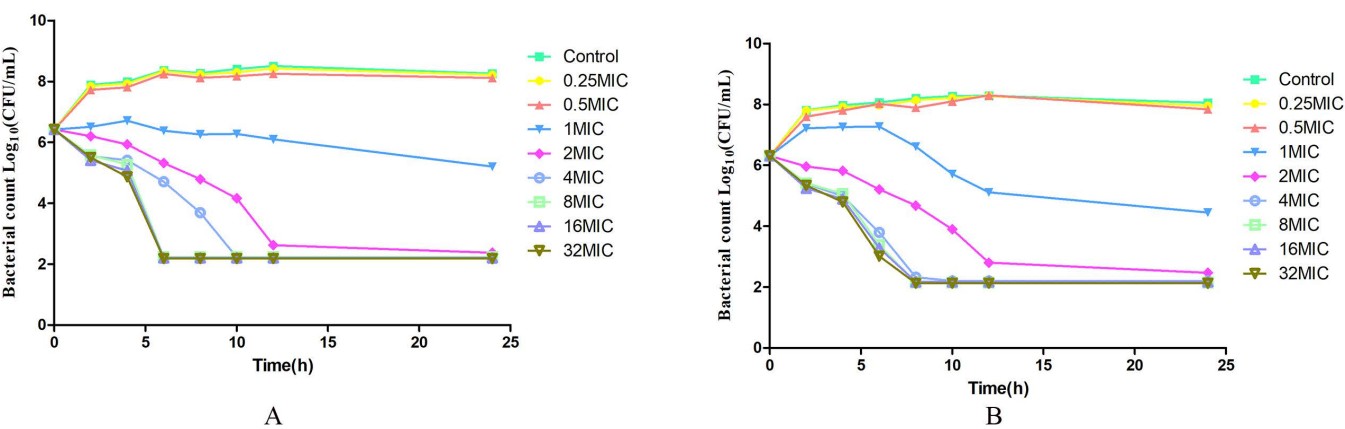

**Fig 6. *In vitro* kill curves of gamithromycin against *Pasteurella multocida* NM-5-7.** A, in MHB; B, in serum.

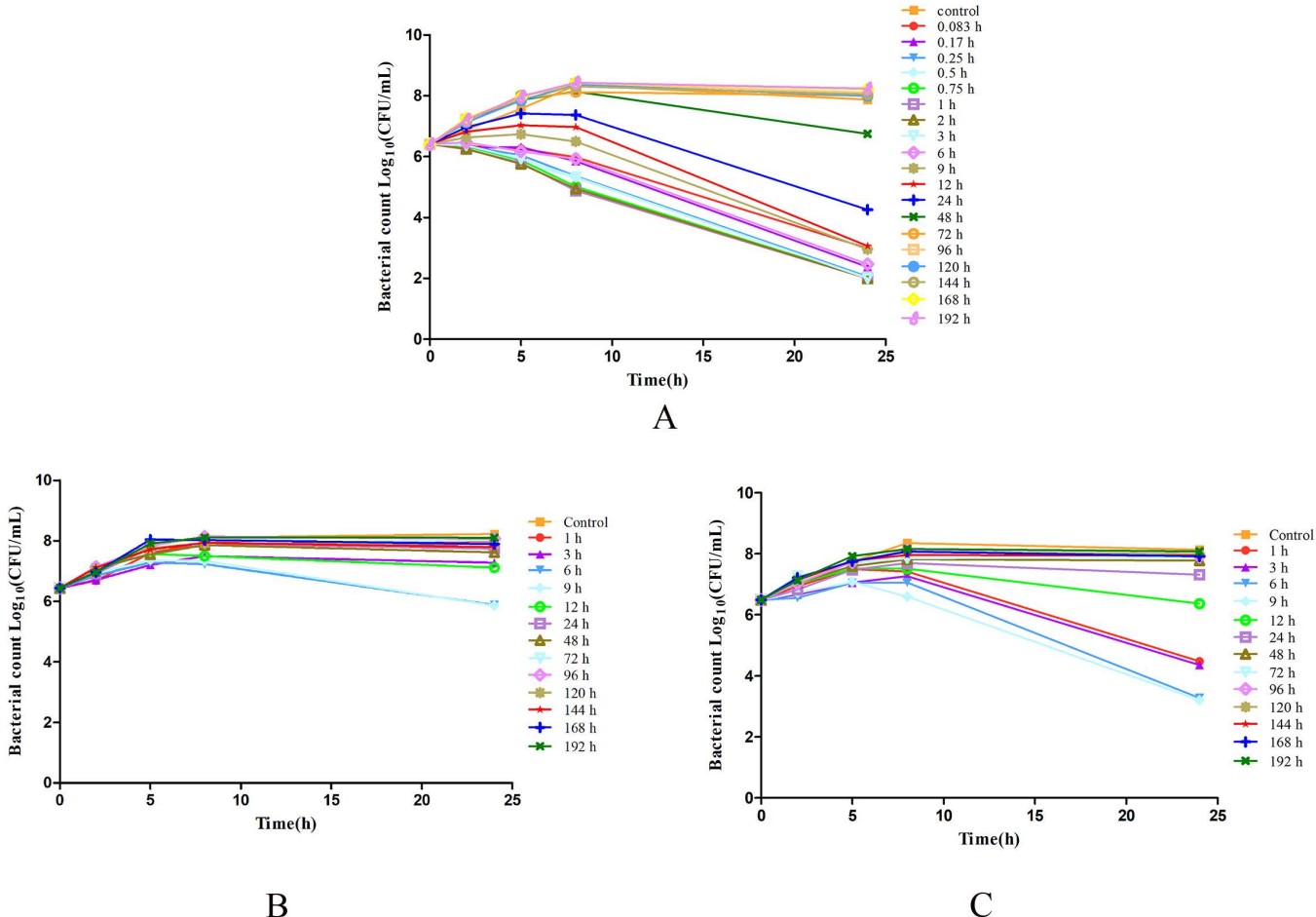

**Fig 7. *Ex-vivo* kill curves of gamithromycin against *Pasteurella multocida* NM-5-7.** A, in serum; B, in transudate; C, in exudate.

obtained at 1, 3, and 12 h showed slight inhibitory activity after 24 h of incubation, while bactericidal effects could be seen in those collected at 6 and 9 h. The exudate samples collected at other time points showed no visible antibacterial activity.

## PK/PD integration and modeling

The inhibitory effect $E_{max}$ associations between *ex-vivo* $AUC_{0-24h}$/MIC values for *P. multocida* NM-5–7 and bacterial counts in the different samples are shown in Fig 8. The PK/PD $AUC_{0-24h}$/MIC values divided by 24 h to evaluate differences in activities are presented in Table 4. In all sample types, the $AUC_{0-24\,h}$/MIC was considered the best PK-PD index for assessing antibacterial activity. As a demonstration of bacteriostatic action, the values obtained after dividing the $AUC_{0-24h}$/MIC by 24 h were 0.27, 0.17, 0.14, respectively, in serum, transudate, and exudate samples. To achieve bactericidal action, the $AUC/MIC_{0-24h}$ divided by 24 h values in serum and exudate were 3.76 and 5.31, respectively, while to achieve bacterial eradication action, the value in serum was 18.46.

## Discussion

Antibiotics are the main methods used for the prevention and treatment of BRD. However, excessive or inappropriate use of antibiotics can result in the development and transmission of drug-resistant strains[21]. The dissemination of

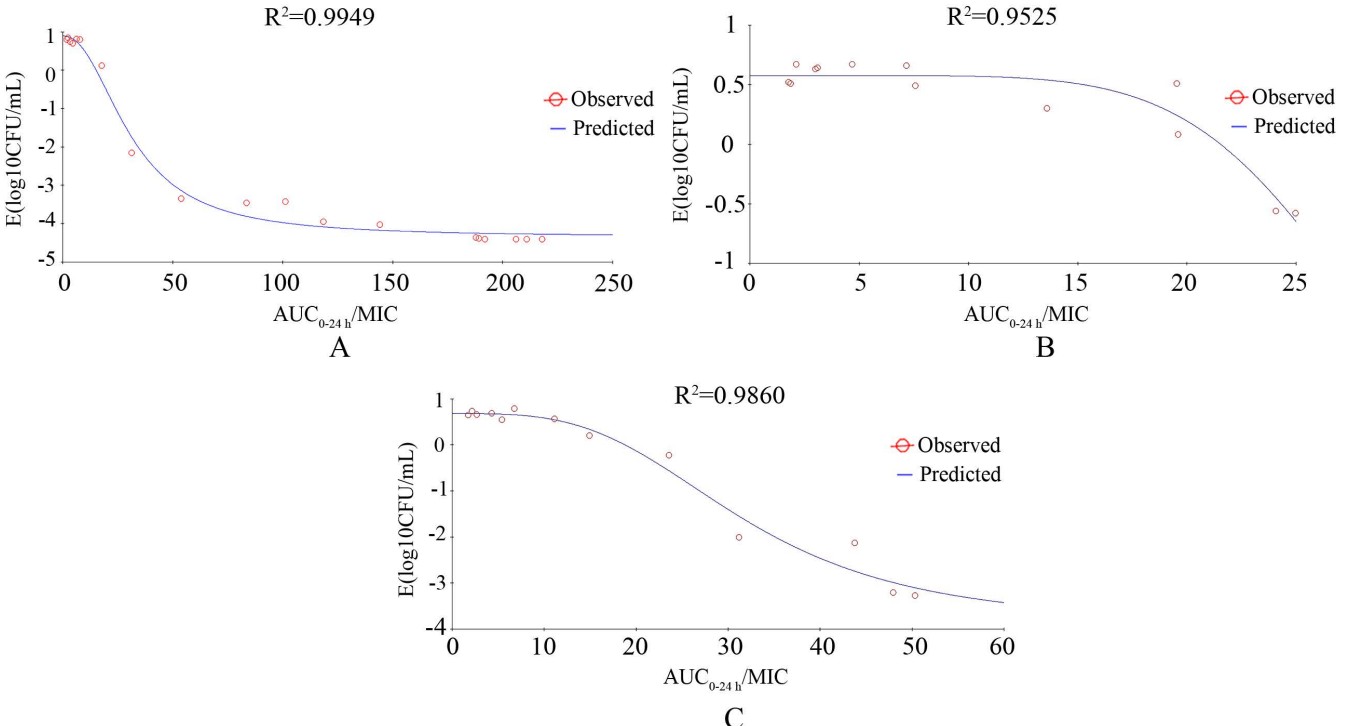

**Fig 8. Correlation between $AUC_{0-24h}$/MIC for *Pasteurella multocida* NM-5-7 and bacterial counts after 24h of treatment.** $R^2$ denotes the correlation coefficient. A, in serum; B, in transudate; C, in exudate.

**Table 4. The PK/PD parameter estimates after division of the $AUC_{0-24h}$/MIC by 24h to assess antibacterial effects.**

| Parameters (units) | Serum | Transudate | Exudate |
|---|---|---|---|
| $E_{max}$ (log$_{10}$ CFU/mL) | 0.89 | 0.58 | 0.68 |
| $EC_{50}$ | 1.30 | 1.24 | 1.32 |
| $E_0$ (log$_{10}$ CFU/mL) | -4.34 | -4.22 | -3.92 |
| Slop (N) | 2.21 | 6.17 | 3.34 |
| $AUC_{0-24h}$/MIC by 24h for bacteriostatic | 0.27 | 0.17 | 0.14 |
| $AUC_{0-24h}$/MIC by 24h for bactericidal | 3.76 | _ | 5.31 |
| $AUC_{0-24h}$/MIC by 24h for eradication | 18.46 | _ | _ |

$E_{max}$ is the corresponding bacterial growth in the absence of drug; $E_0$ is the maximum antibacterial growth inhibition determined as difference in log$_{10}$ CFU/mL in samples over 24-h incubation; $EC_{50}$ is the dividing $AUC_{0-24h}$/MIC by 24h value producing 50% of the maximal antibacterial effect; $N$ is the Hill coefficient

drug-resistant strains in animals used for food not only threatens the health of livestock and poultry but also affects human health[22]. Appropriate antibiotic use is essential to reduce the development of drug resistance and maximize the antimicrobial efficacy of the drugs. The combined use of PK and PD for the prediction of the efficacy of antimicrobial agents and to determine effective dosages is useful for optimizing the dosage regimen[8]. Here, the PK/PD of gamithromycin was investigated using an *ex vivo* tissue cage model. It was found that gamithromycin was rapidly absorbed when delivered at 6mg/kg s.c., but was eliminated slowly. Furthermore, the AUC/MIC data provided an accurate prediction of gamithromycin efficacy

The recommended dose of gamithromycin is 6 mg/kg. After s.c. administration, the $C_{max}$ value (0.43 µg/mL) found here was similar to the previously reported value in sheep (0.45 µg/mL)[13] and pigs (0.41 µg/mL) [15], but lower than the value found for rabbits (1.64 µg/mL)[5], broiler chickens (0.89 µg/mL)[23], and piglets (0.61 µg/mL)[24]. The $t_{1/2\beta}$ (48.30 h) of cattle serum observed here was consistent with previously reported values (42.5 h) in sheep[13], but higher than those in broiler chickens (11.63 h)[23], rabbits (31.5 h) [5], turkeys (34.9 h) [14], and piglets (29.0h)[24]. These discrepancies may be due to species differences. In this study, after s.c. administration, the $AUC_{0-last}$ was 6.23 µg.h/mL, which was higher the observed values in piglets (5.45 µg.h/mL)[24], pigs (3.48 µg.h/mL)[15], turkeys (5.14 µg.h/mL)[14], and broiler chickens (4.09 µg.h/mL)[23], and lower than the value found in sheep (8.88 µg.h/mL)[13]. Gamithromycin has been reported to have high bioavailability (97.6–112%) after s.c. administration[25]. In this study, the bioavailability following s.c. administration was found to be 112.95%, which was similar to the value found in pigs (117.6%) and higher than those reported in broiler chickens (102.4%) and rabbits (86.7%). These discrepancies may be due to species differences.

Effective clinical outcomes and bacterial eradication using macrolides in both human and veterinary medicine often occur with plasma/serum concentrations significantly below the *in vitro* MICs. This disparity between *in vivo* drug levels and MICs has led to frequent questioning of the applicability of traditional PK/PD concepts to macrolides[9]. *Pasteurella multocida* is an exclusively extracellular pathogen, with the pulmonary epithelial lining fluid (PELF) serving as its primary location. The high levels of the drug observed in the PELF have, however, been considered artificially elevated due to the release of the drug caused by cell lysis during bronchoalveolar lavage used for collecting the PELF[26]. Based on the reasons outlined above, a tissue cage model was utilized here for fitting the PK and PD data.

Earlier research has indicated that MIC values determined in MHB may be artificially elevated relative to those seen in other media and fluids[27]. Therefore, MIC values obtained from biological fluids and matrices are preferred over those from MHB. It was found that the MICs in MHB were 16 times greater than those in sera, transudates, and exudates. It is essential to determine MIC values in the biological matrix to ensure the integrity of the integrated PK-PD model in subsequent steps. Higher MIC values obtained from media can result in relatively lower PK-PD indices. Consequently, lower PD target values may prompt dosage reductions during calculations[28]. Based on the current study, the dose of gamithromycin required to achieve a bactericidal effect via subcutaneous administration is 6.04 mg/kg, which is close to the officially recommended dose. However, this dose calculation is based on the MIC value of a single strain rather than the $MIC_{90}$ value. The $MIC_{90}$ represents the minimum drug concentration required to inhibit the growth of 90% of the tested strains and is generally considered a more reliable and clinically representative pharmacodynamic parameter. Since the susceptibility of different strains to the drug may vary significantly, relying solely on the MIC value of a single strain may lead to limitations in dose recommendations. Therefore, future studies should include more clinical isolates to calculate the $MIC_{90}$ value and integrate population pharmacokinetic models to further optimize dosing regimens and ensure their effectiveness in broader clinical scenarios.

In this study, the Vd value after i.v. injection was 54.33 L/kg while that after s.c. injection was 65.73 L/kg/. These values suggest the likelihood of effective tissue penetration. Examination of the timing and degree of the transfer of gamithromycin into the exudates and transudates showed that this was characterized by a relatively slow $T_{max}$, with 7.00 h observed for the exudate and 9.50 h for the transudate. Similar results have been observed for marbofloxacin in a cattle tissue-cage model[20]. The $T_{1/2\beta}$ values in the transudates and exudates were higher than those in the sera. The extensive distribution of gamithromycin and its comparable rate and extent of entry into the exudate and transudate highlight its ability to readily penetrate cell membranes, regardless of inflammatory reactions.

The selection of effective PK/PD indices is critical for optimizing dosage regimens[29, 30]. Here, it was found that the PK/PD index AUC/MIC could accurately predict the effectiveness of gamithromycin against *P. multocida*. However, as the AUC/MIC includes a time dimension (hours), this can present difficulties in comprehension and discussion. To address this, dividing the AUC/MIC by the time period results in a dimensionless ratio, facilitating clearer clinical interpretation of the computed numerical value[11]. In this study, the values obtained by dividing the $AUC_{0-24\,h}$/MIC by 24 h were integrated

with the decrease in bacterial numbers after 24 h incubation using the inhibitory effect $E_{max}$ model. For serum, these values for achieving bacteriostatic, bactericidal, and eradication were 0.27, 3.76, and 18.46, respectively, while for transudates and exudates, the values were 0.17 and 0.14, respectively, lower than those for sera. The PK/PD indices obtained in this study can then be used for dose calculation.

The model has several limitations. First, only a single strain of *P. multocida* was used, and further studies should be conducted using more strains. Second, due to differences between *ex-vivo* and *in vivo* conditions, the predicted dosages should be evaluated in clinical practice. There are many factors apart from PK, PD, and MICs that contribute to the effectiveness of the treatment regimen, including the breed of cattle, age, physical condition, and environmental conditions, which all play crucial roles in disease outcomes. In addition, there is a suggestion for future development to establish antimicrobial dosage schedules based on population PK data to associate the effectiveness of antimicrobial drugs in disease.

In conclusion, the PK/PD index values determined in this study could assist the development of a dosing regimen aimed at achieving bacteriological cure while mitigating the development of resistance. However, these PK/PD index values should be used cautiously. The involvement of other bacteria in BRD, such as *M. haemolytica* and *H. somni*, could affect the efficacy of the dosing regimen. All these factors remain to be evaluated in further studies.

## Supporting information

**S1 Table. *In vitro* killing curve in serum.**
(DOCX)

**S2 Table. In vitro killing curve in MHB.**
(DOCX)

**S3 Table. The gamithromycin concentration in serum after intravenous injection.**
(DOCX)

**S4 Table. The gamithromycin concentration in transudate and exudate samples after intravenous injection.**
(DOCX)

**S5 Table. The gamithromycin concentration in serum after subcutaneous injection.**
(DOCX)

**S6 Table. The gamithromycin concentration in transudate and exudate samples after subcutaneous injection.**
(DOCX)

## Acknowledgments

The authors are grateful to all participating cattle farm and Prof. Yun Liu from Northeast Agricultural University for helpful guidance.

## Author contributions

**Conceptualization:** Qingwen Yang, Xuesong Liu.

**Data curation:** Yushen Li.

**Formal analysis:** Yushen Li.

**Investigation:** Qingwen Yang, Xuesong Liu.

**Writing – original draft:** Yongzhi Lv.

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
