## [Decision Letter · Decision Letter 0]

11 Dec 2024

PONE-D-24-31184Pharmacokinetics/pharmacodynamics of gamithromycin for treating  Pasteurella multocida infection in cattle using a tissue cage modelPLOS ONE

Dear Dr. Liu,

Thank you for submitting your manuscript to PLOS ONE. After careful consideration, we feel that it has merit but does not fully meet PLOS ONE’s publication criteria as it currently stands. Therefore, we invite you to submit a revised version of the manuscript that addresses the points raised during the review process.

We look forward to receiving your revised manuscript.

Kind regards,

Arifa Mehreen, Ph.D.

Academic Editor

PLOS ONE

Journal Requirements:

This study was supported by the Science and Technology Research Program of Chongqing Municipal Education Commission(Grant No.KJZD-K202303501,KJQN-202203511, KJQN-202203504, KJQN-202303507,KJQN-202403510).  

Reviewers' comments:

Reviewer's Responses to Questions

**Comments to the Author**

1. Is the manuscript technically sound, and do the data support the conclusions?

Reviewer #1: Yes

Reviewer #2: Partly

2. Has the statistical analysis been performed appropriately and rigorously? 

Reviewer #1: Yes

Reviewer #2: No

3. Have the authors made all data underlying the findings in their manuscript fully available?

Reviewer #1: Yes

Reviewer #2: Yes

4. Is the manuscript presented in an intelligible fashion and written in standard English?

Reviewer #1: Yes

Reviewer #2: Yes

5. Review Comments to the Author

Reviewer #1: To summarize, the study investigates the PK/PD parameters of gamithromycin for the treatment of BRD by Pasteurella multocida by examining the serum, transudates and exudates of 2 groups of cattle (3 each in a group, two way cross over, by 6mg/kg i.v and .c injection), using tissue cage method. The study shows that overall s.c. injection has higher Cmax, Tmax, T1/2b, Vd and slower clearance compared to i.v. injection.

Minor issue: The authors have not specified how the parameters are expressed in tables 2 and 3 (mean+/-SEM) and n=6 or 3?

Reviewer #2: The pharmacokinetics and pharmacodynamics of the Azalide drug, Gamithromycin has been determined.

1- what is the control/ reference drug used along with gamithromycin to compare all the measured properties.

2- The gamithromycin drug purchased from a particular company, how is that specific company selected and at which basis.

3- The drug concentrations or levels are measured at different time intervals. Is any consideration is given to the fact that inoculum of the microorganism is also standardized at various time intervals. How is it made possible to keep the concentration of the inoculum uniform at various time intervals.

4- the ATCC or other specific number/lot of the Pasteurella multocida is not mentioned. How is it confirmed that the particular strain of Pasteurella multocida is susceptible, resistant or indeterminate?

Overall its a good study, but there is some things important that need to be taken care of, and explained in details.

6. PLOS authors have the option to publish the peer review history of their article (what does this mean? ). If published, this will include your full peer review and any attached files.

**Do you want your identity to be public for this peer review?** For information about this choice, including consent withdrawal, please see our Privacy Policy .

Reviewer #1: No

Reviewer #2: **Yes: ** Dr Zille Huma

---

## [Author Response · Author response to Decision Letter 1]

26 Dec 2024

Dear Reviewers,

Thank you for your letter and for the reviewers' comments concerning our manuscript. Those comments are all valuable and helpful for revising and improving our paper, as well as the important guiding significance to our researches. We have studied comments carefully and have made correction which we hope meet with approval. Revised portion are marked in the paper. The main corrections in the paper and the responds to the reviewer's comments are as flowing:

Reviewer #1: To summarize, the study investigates the PK/PD parameters of gamithromycin for the treatment of BRD by Pasteurella multocida by examining the serum, transudates and exudates of 2 groups of cattle (3 each in a group, two way cross over, by 6mg/kg i.v and .c injection), using tissue cage method. The study shows that overall s.c. injection has higher Cmax, Tmax, T1/2b, Vd and slower clearance compared to i.v. injection.

Minor issue: The authors have not specified how the parameters are expressed in tables 2 and 3 (mean+/-SEM) and n=6 or 3?

Response:

The PK parameters were conducted using WinNonlin software and presented as mean±SD. The n=6.

Reviewer #2: The pharmacokinetics and pharmacodynamics of the Azalide drug, Gamithromycin has been determined.

1- what is the control/ reference drug used along with gamithromycin to compare all the measured properties.

Response:

Thank you for your comment regarding the control/reference drug in our study . We acknowledge that we did not specifically set a traditional control drug in the experiment. However, we compared the measured properties of gamithromycin with those from numerous previously published articles on this drug.

These references provided valuable data on various aspects such as its pharmacokinetic parameters and pharmacodynamic characteristics in different experimental settings and populations. By conducting this comprehensive literature review and comparison, we aimed to place our findings within the context of existing knowledge about gamithromycin and ensure the reliability and relevance of our results.

2-The gamithromycin drug purchased from a particular company, how is that specific company selected and at which basis.

Response: Gamithromycin was developed by Merial in France, and they developed it into an injectable solution. Therefore, the injectable gamithromycin was purchased from Merial in France. Tiaozhan Biotechnology obtained the New Veterinary Drug Registration Certificate of the People's Republic of China for gamithromycin in 2019 and has won the National Science and Technology Progress Award. Therefore, the gamithromycin standard was purchased from this company.

3-The drug concentrations or levels are measured at different time intervals. Is any consideration is given to the fact that inoculum of the microorganism is also standardized at various time intervals. How is it made possible to keep the concentration of the inoculum uniform at various time intervals.

Response:

We determined the concentration of bacteria and plotted the bacterial growth curve through preliminary experiments, and ensured that the inoculum amount in each experiment was standardized by means of these preliminary experiments.

4-the ATCC or other specific number/lot of the Pasteurella multocida is not mentioned. How is it confirmed that the particular strain of Pasteurella multocida is susceptible, resistant or indeterminate?

Response:

Due to our negligence, it was not clearly stated in the article that we used Streptococcus pneumoniae ATCC 49619 as the quality control strain for the MIC test. We have corrected this error in the article.

Yours sincerely,

Xuesong Liu

---

## [Decision Letter · Decision Letter 1]

12 Feb 2025

PONE-D-24-31184R1Pharmacokinetics/pharmacodynamics of gamithromycin for treating  Pasteurella multocida infection in cattle using a tissue cage modelPLOS ONE

Dear Dr. Liu,

Thank you for submitting your manuscript to PLOS ONE. After careful consideration, we feel that it has merit but does not fully meet PLOS ONE’s publication criteria as it currently stands. Therefore, we invite you to submit a revised version of the manuscript that addresses the points raised during the review process.

We look forward to receiving your revised manuscript.

Kind regards,

Arifa Mehreen, Ph.D.

Academic Editor

PLOS ONE

Journal Requirements:

Reviewers' comments:

Reviewer's Responses to Questions

**Comments to the Author**

1. If the authors have adequately addressed your comments raised in a previous round of review and you feel that this manuscript is now acceptable for publication, you may indicate that here to bypass the “Comments to the Author” section, enter your conflict of interest statement in the “Confidential to Editor” section, and submit your "Accept" recommendation.

Reviewer #1: All comments have been addressed

2. Is the manuscript technically sound, and do the data support the conclusions?

Reviewer #1: Yes

3. Has the statistical analysis been performed appropriately and rigorously? 

Reviewer #1: Yes

4. Have the authors made all data underlying the findings in their manuscript fully available?

Reviewer #1: Yes

5. Is the manuscript presented in an intelligible fashion and written in standard English?

Reviewer #1: Yes

6. Review Comments to the Author

Reviewer #1: I congratulate the authors for presenting the study in a detailed manner. The authors have clearly explained the methodology and results of the study and addressed the issues raised. The limitations of the study and the direction for future research have also been explained clearly.

Minor issue: The authors conclude that the lower target PD values will prompt dosage reductions during calculations. (Line 276) It would be helpful, (if possible) if the authors could propose based on the current study findings, a dosing regimen compared to the current regimen, which could be used for a future comparison trial to determine the correct dosing regimen.

7. PLOS authors have the option to publish the peer review history of their article (what does this mean? ). If published, this will include your full peer review and any attached files.

**Do you want your identity to be public for this peer review?** For information about this choice, including consent withdrawal, please see our Privacy Policy .

Reviewer #1: No

---

## [Author Response · Author response to Decision Letter 2]

12 Mar 2025

Dear Reviewer,

Thank you for your letter and for the reviewers' comments concerning our manuscript. Those comments are all valuable and helpful for revising and improving our paper, as well as the important guiding significance to our researches. We have studied comments carefully and have made correction which we hope meet with approval. Revised portion are marked in the paper. The main corrections in the paper and the responds to the reviewer's comments are as flowing:

Reviewer #1: I congratulate the authors for presenting the study in a detailed manner. The authors have clearly explained the methodology and results of the study and addressed the issues raised. The limitations of the study and the direction for future research have also been explained clearly.

Minor issue: The authors conclude that the lower target PD values will prompt dosage reductions during calculations. (Line 276) It would be helpful, (if possible) if the authors could propose based on the current study findings, a dosing regimen compared to the current regimen, which could be used for a future comparison trial to determine the correct dosing regimen.

Response: Based on the current study, the dose of gamithromycin required to achieve a bactericidal effect via subcutaneous administration is 6.04 mg/kg, which is close to the officially recommended dose. However, this dose calculation is based on the MIC value of a single strain rather than the MIC90 value. The MIC90 represents the minimum drug concentration required to inhibit the growth of 90% of the tested strains and is generally considered a more reliable and clinically representative pharmacodynamic parameter. Since the susceptibility of different strains to the drug may vary significantly, relying solely on the MIC value of a single strain may lead to limitations in dose recommendations.

In the future, we will conduct more in-depth research and use MIC90 as an indicator for dose calculation.

Yours sincerely,

Xuesong Liu

---

## [Editor Report · Decision Letter 2]

15 Apr 2025

Pharmacokinetics/pharmacodynamics of gamithromycin for treating  Pasteurella multocida infection in cattle using a tissue cage model

PONE-D-24-31184R2

Dear Dr.  Xuesong Liu 

We’re pleased to inform you that your manuscript has been judged scientifically suitable for publication and will be formally accepted for publication once it meets all outstanding technical requirements.

Kind regards,

Arifa Mehreen, Ph.D.

Academic Editor

PLOS ONE
---

## [Editor Report · Acceptance letter]

PONE-D-24-31184R2

PLOS ONE

Dear Dr. Liu,

I'm pleased to inform you that your manuscript has been deemed suitable for publication in PLOS ONE. Congratulations! Your manuscript is now being handed over to our production team.

Kind regards,

on behalf of

Dr. Arifa Mehreen

Academic Editor

PLOS ONE